# Use of Insect-Derived Chitosan for the Removal of Methylene Blue Dye from Wastewater: Process Optimization Using a Central Composite Design

**DOI:** 10.3390/ma16145049

**Published:** 2023-07-17

**Authors:** Ilham Ben Amor, Hadia Hemmami, Salah Eddine Laouini, Soumeia Zeghoud, Mourad Benzina, Sami Achour, Abanoub Naseef, Ali Alsalme, Ahmed Barhoum

**Affiliations:** 1Department of Process Engineering and Petrochemical, Faculty of Technology, University of El Oued, El Oued 39000, Algeria; ilhambenamor97@gmail.com (I.B.A.); hemmami.h@gmail.com (H.H.); salah_laouini@yahoo.fr (S.E.L.); zsoumeia@gmail.com (S.Z.); 2Renewable Energy Development unit in Arid Zones (UDERZA), University of El Oued, El Oued 39000, Algeria; 3Laboratory of Biotechnology Biomaterials and Condensed Materials, Faculte de la Technologie, University of El Oued, El Oued 39000, Algeria; 4Water, Energy and Environment Laboratory, National School of Engineers of Sfax, University of Sfax, Sfax 3083, Tunisia; mourad_benzina@yahoo.fr; 5Institut Supérieur de Biotechnologie de Monastir ISBM, Monastir 5000, Tunisia; samnaw2001@yahoo.fr; 6NanoStruc Research Group, Chemistry Department, Faculty of Science, Helwan University, Cairo 11795, Egypt; abanoub.nasief@science.helwan.edu.eg; 7Department of Chemistry, College of Science, King Saud University, Riyadh 11451, Saudi Arabia; solme12@hotmail.co.uk

**Keywords:** chitosan extraction, dye removal efficiency, methylene blue, central composite design, natural adsorbents, adsorption parameters, reaction time

## Abstract

Insects are a readily available source of chitosan due to their high reproductive rates, ease of breeding, and resistance to changes in their ecosystem. This study aimed to extract chitosan from several widespread insects: *Blaps lethifera* (CS-BL), *Pimelia fernandezlopezi* (CS-PF), and *Musca domestica* (CS-MD). The study was also extended to using the obtained chitosans in removing methylene blue dye (MB) from wastewater. The source of the chitosan, the initial concentration of MB dye, and the reaction time were chosen as the working parameters. The experiments were designed using a central composite design (CCD) based on the dye removal efficiency as the response variable. The experimental work and statistical calculation of the CCD showed that the dye removal efficiency ranged from 35.9% to 88.7% for CS-BL, from 18.8% to 47.1% for CS-PF, and from 10.3% to 29.0% for CS-MD at an initial MB concentration of 12.79 mg/L. The highest methylene blue dye removal efficiency was 88.7% for CS-BL at a reaction time of 120 min. This indicates that the extraction of chitosan from insects (*Blaps lethifera*) and its application in dye removal is a promising, environmentally friendly, economical, biodegradable, and cost-effective process. Furthermore, the CCD is a statistical experimental design technique that can be used to optimize process variables for removing other organic pollutants using chitosan.

## 1. Introduction

Water pollution caused by hazardous organic pollutant dyes is a global issue resulting from economic and technological advancements. The pollution of water bodies, particularly by dyes, has become an increasingly critical problem [1]. The discharge of dye ions, especially from the dyeing industry, poses a serious threat to both humans and aquatic ecosystems due to their toxicity and persistence in natural waters. Consequently, regulations governing discharges are becoming more stringent [2,3]. The treatment of wastewater containing dyes poses a common challenge, as conventional techniques such as flocculation, coagulation, chemical oxidation, and membrane separation prove ineffective for handling large volumes [4,5]. As alternatives, adsorption and biodegradation have recently been proposed as substitute techniques for decolorizing organic dyes [6]. In particular, adsorption can be applied to a wide range of dye types, making it a versatile method for dye removal. It is effective for both organic and inorganic dyes, including synthetic dyes used in industries such as textiles, paper, and dye manufacturing.

Natural adsorbents, including wheat straws, rye straws, rice straws, peanut husks, corn cobs, Jerusalem artichoke stems, biochar, and biomass, offer advantages for the removal of dyes from wastewater, due to their abundant availability [7]. These biopolymers possess inherent properties that make them effective adsorbents. With their high surface area, abundant functional groups, and electrostatic interactions, they can efficiently bind and remove dye molecules from wastewater [8]. Chitin and chitosan, derived from natural sources such as *crab shells*, *lobster shells*, shrimp exoskeletons, and insects, hold particular promise as affordable adsorbents. Insects, in particular, can provide a sustainable source of chitosan, contributing to waste reduction and maximizing the utilization of insect biomass [9,10]. Chitosan exhibits desirable characteristics such as non-toxicity, excellent bioavailability, biodegradability, and strong adsorption properties. These properties enable its application in various fields, including wound treatment, drug delivery, food packaging, dietary supplements, chelating agents, pharmaceuticals, biomaterials, and water treatment [11,12,13]. Overall, the use of natural biopolymers like chitosan offers a sustainable and versatile approach for the removal of dyes from wastewater.

Methylene blue (MB), a toxic azo dye, must be handled and used with caution due to its adverse effects at high doses. Ingesting large amounts of MB dye can lead to symptoms like nausea, vomiting, diarrhea, and abdominal pain [14,15]. Various adsorbents have proven successful in removing MB dye from wastewater. Activated carbon is widely used due to its large surface area and physical adsorption properties [16]. Clay minerals like bentonite and kaolinite exhibit excellent adsorption capacity due to their layered structure. Zeolites, known for their porous nature, can effectively bind dye molecules [17]. Biomass-based adsorbents such as rice husks and coconut shells possess functional groups that interact with dyes. Magnetic nanoparticles modified with specific groups allow for easy separation using a magnetic field. Polymeric adsorbents like resins and gels can be customized to enhance dye adsorption capacity by tailoring their functional groups or surface properties [18]. Each adsorbent offers unique properties and adsorption mechanisms, making them suitable for different applications and operating conditions. The selection of the most appropriate adsorbent depends on factors such as adsorption capacity, cost, availability, and the specific requirements of the wastewater treatment process.

Chitosan has been investigated for its capacity to remove various organic compounds (dyes, phenolic and pharmaceutical compounds, herbicides, pesticides, drugs, etc.) from wastewater [19], owing to its excellent adsorption capabilities and biodegradability. The effectiveness of chitosan in removing MB dye from wastewater depends on various factors, including the type of chitosan, chitosan dosage, pH, contact time, and dye concentration [18,19,20]. Surprisingly, the application of a central composite design (CCD) has not been explored for the removal of MB dye from wastewater using chitosan. This study aimed to extract chitosan from widespread insects (*Pimelia fernandezlopezi* (CS-PF), *Blaps lethifera* (CS-BL), and *Musca domestica* (CS-MD)) and optimize the processing conditions to achieve efficient removal of MB dye from wastewater using a CCD. The novelty lies in the utilization of chitosan extracted from insects as an adsorbent for removing methylene blue dye. This approach presents a sustainable and renewable source of chitosan, taking advantage of the abundance and rapid reproduction rates of insects. This study further enhances the novelty by employing the CCD for process optimization, which allows for a systematic exploration of various parameters and their interactions.

## 2. Experimental Section

### 2.1. Materials

Sodium hydroxide (NaOH, 97%), hydrochloric acid (HCl, 99%), hydrogen peroxide (H_2_O_2_, 98%), and methylene blue dye (MB, C_16_H_18_ClN_3_S, 82%) were obtained from BioChem Chemophara (Cosne-Cours-sur-Loire, France). Chitosan was extracted from insects (*Pimelia fernandezlopezi*, *Blaps lethifera*, and *Musca domestica*) that were used as a range of local resources. The insects used in this study are widely distributed worldwide and in Algeria. These insects were not endangered or protected species in the field study. The insects were acquired in a dead and dry condition, and no special permission was required to access the site.

### 2.2. Extraction and Characterization of Chitosan

Chitosan was obtained from a variety of insects, including *Pimelia fernandezlopezi* (CS-PF), *Blaps lethifera* (CS-BL), and *Musca domestica* (CS-MD), and extracted using the method of Ilham et al. [18]. In brief, 30 g of each insect powder was added to 1 mol/L HCl for 1 h at 40 °C for demineralization. Deproteinization processes were then carried out for two hours at 80 °C with 1 mol/L NaOH. The resulting polymer was treated with 10 *v*/*v*% H_2_O_2_ for 30 min at 50 °C to decolorize it. The resulting chitin was then deacetylated using 50 *w*/*v*% NaOH at 100 °C for 4 h to obtain chitosan. Finally, the resulting chitosan was dried for 24 h at 50 °C in a vacuum oven. The chitosan was weighed after drying, and its yield was determined using the method below [18]:(1)$$Yield\left(\%\right)=(weight\;of\;dried\;Chitosan,\;g)/(weight\;of\;insect,\;g))\times 100$$

According to a method described by Rdde et al. [21], we vacuum-dried the produced chitosan for 4 h at 650 °C. Ash (%) was calculated using the following formula:(2)$$(W_{1}/W_{2})\times 100$$
where W_1_ and W_2_ are the sample and residue weights, respectively. The moisture content in the produced chitosan was determined using the gravimetric method. To calculate the water mass, the sample was dried in a vacuum oven at 110 °C for 24 h until a constant weight was achieved. The difference between the weight of the wet sample and the weight of the sample after drying was used to determine the water mass, as per the following equation [22]:(3)$$MC\left(\%\right)=((W_{1}-W_{2})/W_{1})\times 100$$
where W_1_ and W_2_ are the weights of the chitosan samples before and after drying, respectively. A modified version of the Wang and Kinsella [23] technique was employed for the experiment. Centrifuge tubes were filled with 0.5 g of chitosan. To disperse the material, the samples were subjected to vortexing at full speed for 1 min with 10 mL of water. The same procedure was repeated using 10 mL of soybean oil instead of water. Following 30 min of shaking the tube’s contents, the mixture was left at room temperature for 12 h. Subsequently, the materials were centrifuged at 3000 rpm for 20 min. After discarding the supernatant, the tube was weighed again to determine the final weight. The WBC and FBC were calculated using the following formulae [23]:(4)WBC%=((water bound, g)/(weight of sample, g))×10
and
(5)FBC%=((Fat bound, g)/(weight of sample, g))×10

### 2.3. Experimental Design

To minimize the number of experiments, a central composite experimental design was used to optimize the effects of the nature of the extracted chitosan, initial dye concentration, and reaction time on the MB dye removal. The experimental design matrix with the different runs, 10 experiments, and the values of each factor can be found in Table 1.

The test design was performed using the statistical software Design–Expert ver.13. The Box–Wilson method was used in this work because of its ability to generate response surfaces that can be used to predict the ideal intended response with fewer experimental runs (CCD). A two-stage factorial design called CCD modified with center and star points allows for the fitting of quadratic polynomial models [24]. In this plan, the independent variables (inputs) are distributed at 3 levels: high level (+1), lower level (−1), and center points (0) [25]. The values of each independent variable are given in Table 1 in both coded and real numbers. Equation (1) can be used to determine the total reduced number of runs required for this study based on the CCD:(6)$$N=2^{n}+2n+n_{c}=22+2*2=10$$
where *n* is the number of independent variables (two in our case), and *N* is the total number of experimental runs; 2*n*, 2*^n^*, and *n_c_* are assigned to axial, factorial, and center runs, respectively. The recommended number of center points for two variables in Aquert error calculations is four [26].The total number of runs required for this study with *n_c_* = 2 and *n* = 2 was 10, according to Equation (2). The runs performed are listed in Table 2, along with their actual values.

The quadratic model in the central composite design with 2 components can be written as follows:(7)$$Y=\beta_{0}+\sum_{j=1}^{m}\beta_{j}x_{j}+\sum_{j=1}^{m}\beta_{jj}{x_{j}}^{2}+\sum \sum_{i<j=2}^{m}\beta_{ij}x_{i}x_{j}+\varepsilon$$
where $x_{i}\;\mathrm{and}\;x_{j}$ are factors (*i* and *j* vary from 1 to m), and ε illustrates the errors. The coefficients 𝛽*_i_*, 𝛽*_ii_*, and 𝛽*_ij_* represent the linear, combined, and binominal effects, respectively. $\beta_{0}$ is a constant coefficient, and m represents the sum of the factors studied [27,28].

In the equilibrium experiments, a measured amount of chitosan (1 g) was added to a beaker containing 30 mL of aqueous MB dye solution. The initial concentrations of the MB dye solution ranged from 6.39 to 12.79 mg/L. The duration of the experiments varied from 10 to 120 min, depending on the specific time interval. Throughout the experiments, the temperature was maintained at 25 °C, and the mixture was shaken using an orbital shaker at 150 rpm. To halt the adsorption process, a centrifuge operating at a speed of 6000 rpm was utilized to separate the chitosan from the solution. The initial and final concentrations of the MB dye were measured using a UV–vis spectrophotometer (UV-2450, Shimadzu, Kyoto, Japan) at 663 nm, allowing for the determination of the adsorption efficiency for each experiment. The pH of the experimental solution was adjusted to pH = 7 using 1 mol/L HCl and 1 mol/L NaOH as needed. The difference between the concentrations of MB dye in the aqueous solution before and during adsorption was used to estimate the amount adsorbed (*q_e_*) [29]:(8)$$q_{e}=\frac{\left(C_{0}-C_{e}\right)V}{m}$$
where *V* is the volume of the dye solution (L), *m* is the weight of the chitosan (g), and *q_e_* is the dye’s equilibrated concentration on the adsorbent (mg·g^−1^). The initial and equilibrium dye solution concentrations are *C*_0_ and *C_e_* (mg·L^−1^), respectively. The following equation was used to determine the dye removal capacity (%):(9)$$Dye\;removal\left(\%\right)=\frac{\left(C_{0}-C_{e}\right)}{C_{0}}\times 100$$
where *C_e_* and *C*_0_ are the equilibrium and initial dye concentrations (mg·L^−1^), respectively.

## 3. Results and Discussion

### 3.1. Characteristics of Chitosan

The extraction of chitosan was carried out in three main steps, i.e., demineralization, deprotonation, and deacetylation, leading to the isolation and then deacetylation of chitin to obtain chitosan. The main properties of the chitosan are summarized in Table 2. The results show that the dry weight (yield %) of the extracted chitosan was 41.6%, 50.0%, and 57.9% (CS-PF, CS-BL, and CS-MD, respectively). These chitin yields are within or above the average ranges reported for chitin from *cicada sloughs* (36.6%) [30], *crab shells* (10.83%) [31], and beetles (5%) [23]. These results indicate an efficient extraction process that removes proteins and impurities during deacetylation and precipitation compared with previous studies [18].

Commercial chitosan may be impacted by moisture absorption during storage, because chitosan is naturally hygroscopic. The moisture content of chitosan can vary depending on a variety of factors, such as the source of the chitin, the degree of deacetylation, and the method of processing (e.g., temperature, humidity, and exposure to air). Commercial chitosan has a moisture content of about 10% under ambient conditions. The moisture content of the extracted chitosans (CS-PF, CS-BL, and CS-MD) was 17.2%, 14.3%, and 7.8%, respectively. These results indicate higher amounts of dry chitosan isolated from insects (CS-MD) compared to CS-PF, and CS-BL, indicating good-quality chitosan.

The ash content of chitosan is a crucial factor that affects its quality. The moisture content results indicated that the insect-isolated chitosan samples (CS-PF and CS-BL) had lower moisture contents of 2.0% and 1.6%, respectively, compared to CS-MD, which had a higher moisture content of 8.2%. Based on these findings, it can be inferred that the insect-isolated chitosan samples (CS-BL and CS-PF) have higher quality compared to CS-MD and are suitable for biomedical applications. On the other hand, CS-MD may be more suitable for environmental applications [18].

Chitosan exhibits low solubility in water and most organic solvents at neutral and high pH, due to the strong inter- and intramolecular hydrogen bonds between the macromolecular chitosan chains [32]. The water-binding capacity (WBC) was 287.1%, 515.5%, and 301.1% for the chitosans isolated from insects (CS-PF, CS-BL, and CS-MD, respectively). These results indicate a higher WBC of insect-isolated chitosan (CS-BL) compared to CS-PF and CS-MD. The higher WBC values observed for CS-BL and CS-MD indicate that these insect-isolated chitosan samples have a greater capacity to bind and hold water compared to CS-PF. This characteristic can be beneficial in dye removal, where water retention or moisture control is desired.

Chitosan is a natural fibrous polymer with high binding capacity for fats, dyes, oils, and lipids. The binding capacity of chitosan can be affected by a number of factors, such as the degree of deacetylation, the molecular weight, the pH of the solution, and the concentration of the target molecule. In this study, the fat-binding capacity (FBC) was 296.8%, 433.4%, and 455.2% for the insect-isolated chitosans (CS-BL, CS-PF, and CS-MD, respectively). This suggests that CS-MD has a greater ability to bind and retain fats, which can be advantageous in applications where the adsorption of fats is desired [18].

DD is a measure of the extent to which chitin has been converted to chitosan through the removal of acetyl groups [33]. Higher DD values are often associated with enhanced properties such as increased antimicrobial activity or better gelation capabilities. In the case of chitosan isolated from insects (CS-PF, CS-BL, and CS-MD), the provided DD values were 88.3%, 87.1%, and 84.1% respectively. These values indicate the extent of deacetylation for each chitosan sample. Based on these results, it can be inferred that the chitosan isolated from insects exhibits a relatively high DD, which suggests the potential for enhanced properties and performance in various applications.

Analyzing the SEM images of CS-BL (Figure 1a), it can be observed that the chitosan particles have an elongated and fibrous nature, forming an interconnected network-like structure. The surface of CS-BL appears fibrous, with visible strands and intertwining patterns, indicating a fibrous morphology. Moving on to CS-PF (Figure 1b), the SEM image reveals elongated and fibrous chitosan particles, although with a lesser degree of interconnectivity compared to CS-BL. The surface of CS-PF is relatively smooth, displaying fewer visible pores or irregularities. The particles also appear more dispersed and individualized compared to CS-BL. In the case of CS-MD (Figure 1c), the SEM image shows chitosan particles with spherical and rod-like shapes. This indicates a different morphological structure compared to CS-BL and CS-PF. The fibrous morphology of CS-BL and CS-PF, along with their interconnected network-like structures, may contribute to enhanced adsorption capabilities, while the spherical and rod-like shapes of CS-MD may have different effects on its performance in various applications.

### 3.2. Analytical Statistics and Model Fitting

Chitosan’s large amounts of amino functionalities provide unique adsorption properties for numerous organic dyes. The polycationic properties of chitosan are anticipated to promote charged interactions with MB molecules [34]. Covalent bonding, electrostatic interaction, and hydrogen bonding are three ways that MB dye might bond with chitosan’s functional groups. A higher degree of acetylation, grafting (i.e., adding functional groups), or crosslinking processes with other polymers might alter the chitosan molecule, improving its potential to bind harmful contaminants in wastewater and enhancing its tolerance to adverse medium conditions. Because chitosan’s adsorption capacity is high when the value of DD is increased, the DD of chitosan is significant. In previous studies [18,35], it was found that the adsorption capacity increased with increasing DD. As the CS-PF chitosan had the highest DD (88.2%), its adsorption capacity over time was higher than that of CS-BL and CS-MD, which explains how the different sources of chitosan affect the adsorption properties of the dye. 

For MB dye removal, Table 3 shows the statistical combinations of the factors studied and the greatest observed and expected responses.

The experimental work and statistical calculation show that the dye removal efficiency ranges from 35.9% to 88.7% for CS-BL, from 18.8% to 47.1% for CS-PF, and from 10.3% to 29.0% for CS-MD. The highest MB removal efficiency was 88.7% for CS-BL at an initial MB concentration of 12.79 mg/L MB and reaction time of 120 min. The observed differences in MB removal efficiency between CS-BL, CS-PF, and CS-MD can be attributed to various factors, including the chitosan samples’ DD, moisture content, and water-binding capacity (see Table 2) [36,37]. Firstly, the DD of chitosan plays a crucial role in its adsorption capacity. CS-BL and CS-PF had higher DD values (87.1% and 88.3%, respectively) compared to CS-MD (84.1%), indicating a greater number of amino groups available for interaction with dye molecules. This higher DD value could explain the higher MB removal efficiency observed with CS-BL and CS-PF. Secondly, moisture content can influence the adsorption properties of chitosan. CS-BL exhibited a higher moisture content (17.2%) compared to CS-PF (14.3%) and CS-MD (7.8%). As a result, the increased pore size promoted greater dye adsorption capacity, improving the efficiency of MB dye removal. The increased moisture content in CS-BL may have facilitated better swelling and improved accessibility of the chitosan matrix, leading to enhanced interaction with dye molecules and higher MB removal efficiency. Lastly, the water-binding capacity of chitosan is important in creating a hydrated environment that promotes dye adsorption. CS-BL had a higher water-binding capacity (515.5%) compared to CS-PF (287.1%) and CS-MD (301.1%), suggesting that it could provide more favorable conditions for the dye removal process, resulting in higher efficiency. Overall, the combined influence of DD, moisture content, and water-binding capacity contributes to the variations in MB removal efficiency observed between CS-BL, CS-PF, and CS-MD.

The analysis showed that the equation for quadratic polynomial models linking the responses with the studied factors is given in terms of coded factors. The following models are provided for the three responses in percentages (%): Y(CS-PF), Y(CS-BL), and Y(CS-MD), depending on the contribution of each of the parameters (**A** and **B**) in a linear or quadratic manner (**A**^2^ and **B**^2^) to the response Y (%), as well as the possible **AB** interactions.
(10)$$Y\left(CS-PF\right)\%=61.76+6.75A+18.88B-0.7525AB-1.11A^{2}+1.83B^{2}$$
(11)$$Y\left(CS-BL\right)\%=23.83+4.33A+9.52\;B+0.4850\;AB+0.0950\;A^{2}+0.3150\;B^{2}$$
(12)$$Y\left(CS-MD\right)\%=36.04+14.13A+10.73B+1.83\;AB+5.54A^{2}+0.2780B^{2}$$

Statistical parameters were estimated using analysis of variance (ANOVA). The determination coefficient, R^2^, was used to describe how well the experimental findings fit the polynomial model equation. Within a 95% confidence interval, the significance of each term in the polynomial equation was estimated using the F-test. The agreement of the developed model with the experimental results is indicated by the standard deviation (6.0) and the determination coefficient R^2^.

The results of the ANOVA are shown in Table 3, determining the relationships between the dependent factors, the response (MB distance), and the values of R^2^adj, R^2^, adequate precision, lack of fit, F-value, CV, and SD. High F-values and low *p*-values represent the significance of the model performed by the ANOVA test [38]. The R^2^ coefficient, which measures the fit of the model and is defined as the ratio of the specified variable to the total variance, was convincingly good for the model (CS-PF, R^2^: 0.9916; R^2^ Fit: 0.9810; CS-BL, R^2^: 0.9971; R^2^ Fit: 0.9937; CS-MD, R^2^: 0.9915; R^2^ Fit: 0.9809).

The signal-to-noise ratio, represented as adequate precision (AP), measures how well the model can navigate the design space [39]. When AP is larger than 4, the model is deemed to be interesting. As mentioned earlier, the AP values of CS-PF, CS-BL, and CS-MD for MB distance (study response) were equal to 31.2851, 52.4867, and 31.2851, respectively, which are good and acceptable values. The standard deviation values (2.05, 0.6818, and 2.0514 for CS-PF, CS-BL, and CS-MD, respectively) show that the results are significant and very reliable. The low values of CV indicate the high reliability and excellent precision of the experiments. The relationship between Fisher’s *p*-value, *F*-value, and term effect can be used to determine the influence of the studied conditions on the MB dye removal of chitosan. The *p*-value and *F*-value are inversely related; the lower the *p*-value, the higher the *F*-value and the greater the significance of the term effect on the response, and vice versa [40]. Moreover, all linear effects have a *p*-value < 0.05, indicating a significant linear effect for the main parameters (Table 4).

In a CCD, predicted vs. experimental values plots are used to evaluate the accuracy and validity of the mathematical model that describes the relationship between the process factors and the response variable. Figure 2 shows the plots of the expected and actual values for the MB dye removal. The predicted vs. experimental values plot shows a straight line with a slope of one, indicating perfect agreement between the predicted and experimental values and demonstrating the efficiency of the models in prediction.

### 3.3. Internal vs. Normal

In a CCD, the normal probability plot is used to check the assumption that the response variable follows a normal distribution. If the response variable is normally distributed, the normal probability plot will show a straight line, while deviations from a straight line indicate non-normality. The linear pattern shown in Figure 3 indicates when the data are regularly distributed, and vice versa. In this instance, it shows that the data are regularly distributed, because they are widely scattered on a straight line [41].

### 3.4. Residual Analysis

Residual analysis is an important step in the CCD to evaluate the quality of the experimental data and to check whether there are deviations from the assumptions of the statistical model. The residuals are the difference between the observed and predicted values of the response variable (dye removal) and are a measure of the quality of the fit of the CCD model to the data. Figure 4 shows the residuals compared to the run number plot, which is often referred to as the outlier t plot. The red lines above and below the figure correspond to the 95% confidence interval of the control range. In this case, all points are within the acceptable range where the residual points should be (i.e., between the two red lines). Also, the graph shows randomly distributed data points with no discernible pattern or shape, indicating that there is no serial correlation and that the residuals are normally distributed.

### 3.5. Process Factors’ Impact on MB Removal

In the CCD, the process factors are deliberately manipulated based on a predetermined plan—often a response surface design—to analyze their impact on the response variable of interest. This design includes central points and star points, which facilitate the estimation of linear, quadratic, and interaction effects of the process factors on the response variable. Visualizing the data using 2D and 3D plots helps in understanding the relationships between the variables and the response. Figure 5 presents contours and response surface plots, offering a graphical representation of the variations in methylene blue (MB) removal using CS extracted from three different insects. The contours demonstrate that higher MB removal is achieved at higher MB concentrations and longer reaction times. Notably, CS-BL exhibits the highest MB removal efficiency, followed by CS-MD and CS-PF, indicating that the extraction of chitosan from insects—particularly CS-BL—shows promising potential for dye removal applications.

### 3.6. Optimization by the Desirability Functions

Desirability functions play a crucial role in assessing the predicted response with the highest possibility, as determined by previous studies [42]. Throughout the optimization process, the desirability function was employed, ranging from 0 (undesirable) to 1 (desirable), to evaluate the desirability of different factors. The numerical optimization approach was utilized to identify the maximum values of the desirability function. In this study, desirability functions were utilized to optimize the removal of MB dye. The Design–Expert 13 software facilitated the setting of optimization goals, allowing for the maximization, minimization, or attainment of desired values within specific response ranges. The primary optimization criterion was to achieve the highest MB removal, while keeping other factors within their minimum (−1) and maximum (+1) ranges. A maximum desirability of 1.00 was obtained as the result of mathematical analysis. Figure 6 illustrates the optimization graph, displaying the best predicted values for the maximum removal percentage of MB and the corresponding desirability function. Under these optimized conditions, the maximum MB removal (%) for CS-PF (with an experimental value of 67.17%) was achieved at an MB concentration (A) of approximately 10.31 mg/L and a reaction time (B) of around 75.77 min. Similarly, CS-BL achieved a maximum MB removal (%) of 32.37% (experimental value) at an MB concentration (A) of approximately 10.75 mg/L and a reaction time (B) of approximately 103.54 min. Lastly, CS-MD achieved a maximum MB removal of 33.6% (experimental value) at an MB concentration (A) of approximately 9.05 mg/L and a reaction time (B) of approximately 63.86 min.

## 4. Mechanism of Interaction and MB Dye Removal

Chitosan, known for its rich presence of three primary functional groups, exhibits remarkable adsorption capabilities toward a wide range of water pollutants (e.g., metal ions and organic dyes) [18]. Firstly, chitosan possesses an amino group (-NH_2_) as its primary functional group. The -NH_2_ groups in chitosan contain a lone pair of electrons, which can be also used to form coordination bonds with positively charged ions. Secondly, chitosan contains numerous hydroxyl groups (-OH) attached to the sugar units, which engage chitosan in hydrogen bonding with other functional molecules. The presence of -OH and -NH_2_ groups enhances the hydrophilic nature of chitosan, enabling it to dissolve in water and form hydrogen bonds with functional molecules. Lastly, chitosan incorporates acetyl groups (-COCH_3_) attached to the amino groups. These contribute to the overall structure of chitosan and further enhance its adsorption properties toward MB dye [18]. 

MB contains a thiazine ring, which is a heterocyclic ring containing both nitrogen (N) and sulfur (S) atoms. This thiazine ring is fused to the planar aromatic ring system known as the phenothiazine moiety. The phenothiazine ring consists of alternating single and double bonds, allowing for the delocalization of π-electrons across the conjugated system. The thiazine ring and the phenylamino group, which is attached to the thiazine ring, act as prominent chromophore groups in MB. The characteristic blue coloration of MB dye can be attributed to the presence of these chromophore groups. These chromophores enable the absorption of light in the red–orange region of the electromagnetic spectrum. The presence of the attached dimethylamino group (-N(CH_3_)_2_) in MB does not directly contribute to the color of MB. However, it does play a crucial role in enhancing MB’s reactivity and electron-donating properties. The -N(CH_3_)_2_ group strengthens MB’s capacity to engage in hydrogen bonding with other molecules. This enhanced interaction is a result of the strong electron-donating nature of the -N(CH_3_)_2_ group, which is attributed to the attachment of two methyl (-CH_3_) groups to the amino nitrogen (-NH-) atom.

Drawing from the preceding discussion, the adsorption mechanism of MB dye by chitosan can be elucidated through three primary modes of interaction, as depicted in Figure 7: (1) Hydrophobic and π–π stacking interactions: The hydrophobic regions (e.g., -CH_3_ groups) of the dye molecules can interact with the hydrophobic regions of the chitosan chains (e.g., -CH_3_ groups) through hydrophobic interactions.

This interaction can result in the adsorption or binding of the MB molecules to the chitosan surface or within its structure. Additionally, the aromatic rings present in MB can undergo π–π stacking interactions with the aromatic rings of chitosan. (2) Electrostatic interactions: MB contains a positively charged group (S^+^), which can interact with the negatively charged groups on chitosan (-COCH_3_), leading to electrostatic attractions between the two molecules [43]. (3) Hydrogen bonding: MB dye contains nitrogen atoms in its structure, which can act as hydrogen-bond acceptors. Chitosan, on the other hand, contains -OH and -NH_2_ groups that can act as hydrogen-bond donors and acceptors. These three ways of interaction (hydrophobic interactions, electrostatic attractions, and hydrogen bonding) collectively explain the bonding and uptake of MB dye by chitosan. It is important to note that the actual adsorption mechanism can be influenced by various factors, such as pH, temperature, dye concentration, and the specific properties of the chitosan used.

Lastly, it is important to highlight that the focus of this particular study was primarily on investigating the adsorption capacity of chitosan for removing MB dye, and the process of regenerating chitosan was not explored. To reuse chitosan for future dye removal, regeneration becomes necessary, which involves removing the adsorbed dye molecules from the surface of the chitosan. Various regeneration methods are available, such as adjusting the pH of the solution, using organic solvents, or employing specific desorbing agents. The choice of method depends on the adsorption conditions and the desired regeneration efficiency. However, the regeneration aspect was beyond the scope of this study, and further research or experimentation would be necessary to evaluate the effectiveness and feasibility of regenerating chitosan for subsequent dye removal.

## 5. Conclusions

Chitosan is a modified natural carbohydrate polymer that is found in a variety of natural sources, including crustaceans, mollusks, fungi, and insects. Chitosan from insects is readily available due to their reproductive rate, ease of cultivation, and resistance to changes in their ecosystem. The production of chitosan from insects typically involves the following steps: First, the insects are collected and processed to remove any unwanted materials, such as legs and wings. The chitin is then extracted from the exoskeleton using a combination of mechanical and chemical processes. Finally, the chitin is deacetylated to produce chitosan. In this research, chitosan was extracted from several insects (*Blaps lethifera* (CS-BL), *Pimelia fernandezlopezi* (CS-PF), and *Musca domestica* (CS-MD)) and then used to remove methylene blue (MB) dye from an aqueous solution. Our results showed that insect chitosan has similar properties to chitosan extracted from other sources, such as yield (41.5–57.2%), moisture content (7.8–17.2%), ash contents (1.6–8.2%), water-binding capacity (287.1–515.5%), fat-binding capacity (296.8–455.2%), and degree of deacetylation (84.1–88.3%). However, the use of insect chitosan is still in its early stages, and further research is needed to fully understand its properties and potential applications. In addition, concerns about the safety and ethics of using insects for industrial purposes may need to be addressed before insect chitosan becomes a widely accepted alternative to crustacean-derived chitosan. For wastewater treatment, a CCD was used as an effective statistical technique to study the effects of MB removal using significant independent variables such as sources of chitosan, initial MB concentration, and reaction time. The results showed that of the factors tested, the chitosan source and MB concentration have the greatest effect on MB removal, followed by the contact time. The experimental work and statistical calculation showed that the dye removal efficiency ranged from 35.9% to 88.7% for CS-BL, from 18.8% to 47.1% for CS-PF, and from 10.3% to 29.0% for CS-MD, at an initial MB concentration of 12.79 mg/L. It can be concluded that chitosan can be used as a natural substitute for ingredients such as activated carbon, which are commonly used to remove dyes from wastewater. Overall, insect chitosan represents a promising and sustainable biopolymer for various applications that require biodegradable, biocompatible, and antimicrobial materials.

## Figures and Tables

**Figure 1 materials-16-05049-f001:**
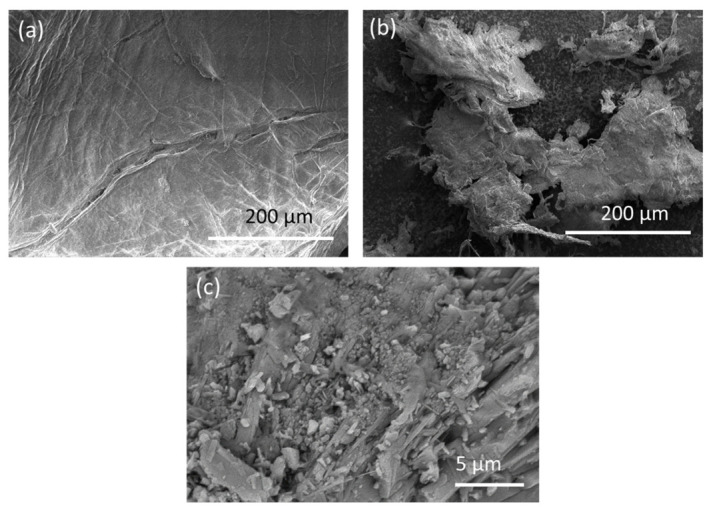
SEM analysis of the chitosans: (**a**) *Blaps lethifera* (CS-BL), (**b**) *Pimelia fernandezlopezi* (CS-PF), and (**c**) *Musca domestica* (CS-MD).

**Figure 2 materials-16-05049-f002:**
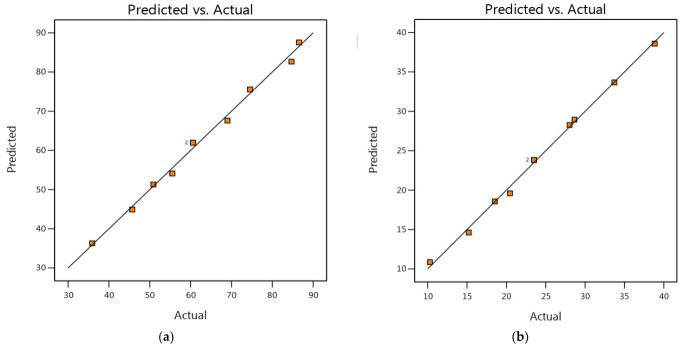
Predicted vs. experimental values for MB dye removal using insect-derived chitosans: (**a**) *Blaps lethifera* (CS-BL), (**b**) *Pimelia fernandezlopezi* (CS-PF), and (**c**) *Musca domestica* (CS-MD).

**Figure 3 materials-16-05049-f003:**
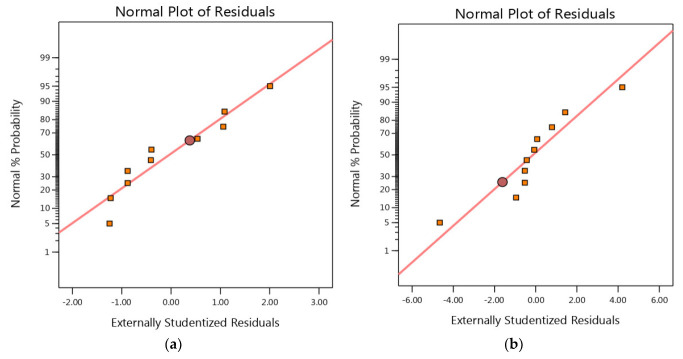
Normal probability distribution of residuals of MB dye for insect-derived chitosans: (**a**) *Blaps lethifera* (CS-BL), (**b**) *Pimelia fernandezlopezi* (CS-PF), and (**c**) *Musca domestica* (CS-MD).

**Figure 4 materials-16-05049-f004:**
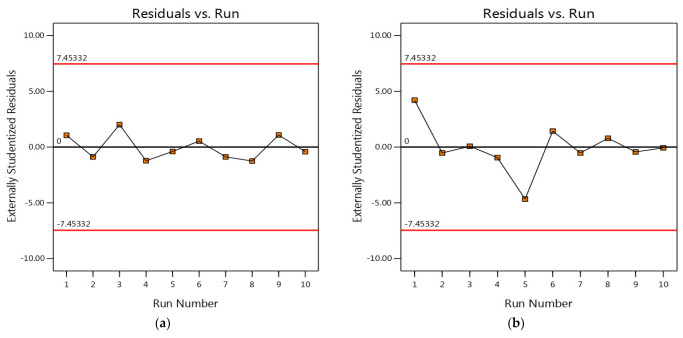
The externally studentized residuals and run numbers for MB dye removal using insect-derived chitosans: (**a**) *Blaps lethifera* (CS-BL), (**b**) *Pimelia fernandezlopezi* (CS-PF), and (**c**) *Musca domestica* (CS-MD).

**Figure 5 materials-16-05049-f005:**
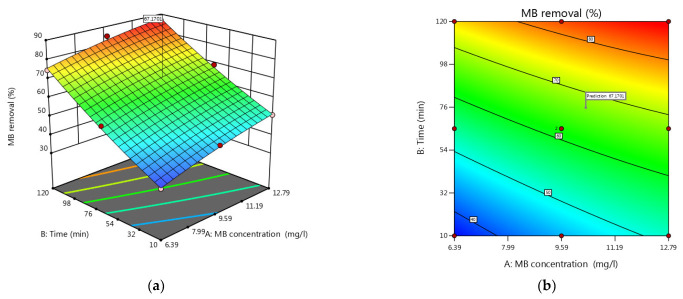
Contours and response surface plots for MB removal with insect-derived chitosans: (**a**,**b**) *Blaps lethifera* (CS-BL), (**c**,**d**) *Pimelia fernandezlopezi* (CS-PF), and (**e**,**f**) *Musca domestica* (CS-MD).

**Figure 6 materials-16-05049-f006:**
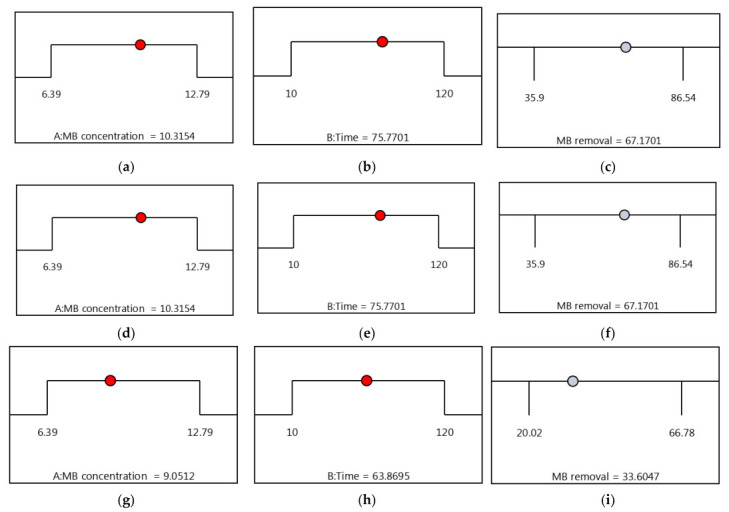
Desirability ramp for optimization of maximal elimination of MB using insect-derived chitosans: (**a**–**c**) *Blaps lethifera* (CS-BL), (**d**–**f**) *Pimelia fernandezlopezi* (CS-PF), and (**g**–**i**) *Musca domestica* (CS-MD).

**Figure 7 materials-16-05049-f007:**
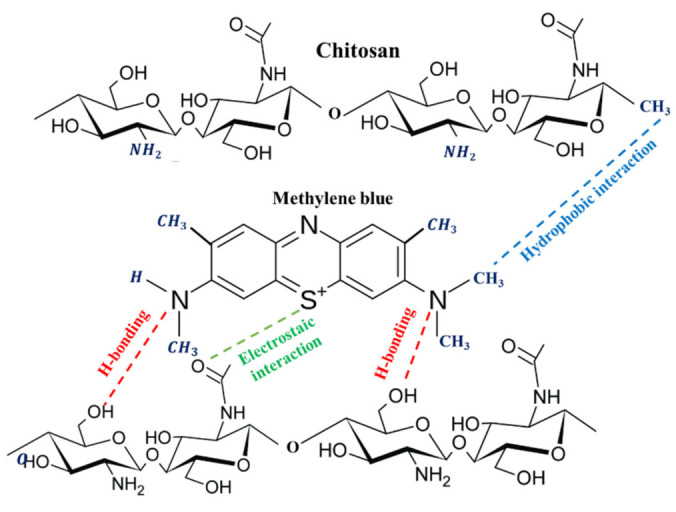
Schematic diagram showcasing the hydrophobic interactions, electrostatic attractions, and hydrogen bonding between chitosan and MB dye.

**Table 1 materials-16-05049-t001:** The various levels and codes of the factors under study.

Parameters	Codes	Level		
		−1.0	0.0	1.0
MB concentration (mg/L)	A	6.39	9.59	12.79
Time (min)	B	10.00	65.00	120.00

**Table 2 materials-16-05049-t002:** Properties of chitosan extracted from *Blaps lethifera* (CS-BL), *Pimelia fernandezlopezi* (CS-PF), and *Musca domestica* (CS-MD). The values are based on at least 3 measurements.

Characteristics of Chitosan	Sources of Chitosan Extracts		
	CS-BL	CS-PF	CS-MD
Yield (Y)	41.5 ± 0.5%	50.3 ± 0.3%	57.9 ± 0.2%
Moisture content (MC)	17.2 ± 0.2%	14.3 ± 0.3%	7.8 ± 0.1%
Ash content (AC)	1.6 ± 0.1%	2.1 ± 0.1%	8.2 ± 0.2%
Water-binding capacity (WBC)	515.5 ± 6.5%	287.1 ± 5.8%	301.1 ± 4.3%
Fat-binding capacity (FBC)	433.4 ± 11.3%	296.8 ± 14%	455.2 ± 13.2%
Degree of deacetylation (DD)	87.1 ± 0.2%	88.3 ± 0.1%	84.1 ± 0.3%
Crystallinity index	84 ± 0.1%	73 ± 0.4%	81 ± 0.2%

**Table 3 materials-16-05049-t003:** Design matrix for experimental responses and factors at various levels of factors.

Experiment Order	Experimental Factors		Response (MB Removal)					
	MB Concentration (mg/L)	Time (min)	Actual			Predicted		
			CS-BL	CS-PF	CS-MD	CS-BL	CS-PF	CS-MD
1	6.39	10	35.9	10.3	20.0	36.3	10.9	18.8
2	12.79	10	50.9	18.5	43.6	51.3	18.6	43.4
3	6.39	120	74.6	28.7	35.9	75.6	29.0	36.6
4	12.79	120	86.5	38.8	66.8	87.6	38.6	68.5
5	6.39	65	55.5	20.5	27.0	54.1	19.6	27.5
6	12.79	65	69.0	28.0	57.3	67.6	28.3	55.7
7	9.59	10	45.7	15.2	24.3	44.9	14.6	25.6
8	9.59	65	60.6	23.5	36.0	62.0	23.8	36.0
9	9.59	120	84.7	33.7	49.5	82.7	33.6	47.1
10	9.59	65	60.6	23.5	36.0	62.0	23.8	36.0

**Table 4 materials-16-05049-t004:** Adsorption using ANOVA: ^a^ significant; ^b^ not significant.

**Source (CS-PF)**	**Sum of Squares**	**Degree of Freedom**	**Mean Square**	***F*-Value**	***p*-Value**
Model	2424.35	5	484.87	128.39	0.0002 ^a^
A	273.65	1	273.65	72.46	0.0010
B	2139.10	1	2139.10	566.41	<0.0001
AB	2.27	1	2.27	0.5998	0.4819
A^2^	2.87	1	2.87	0.7603	0.4325
B^2^	7.78	1	7.78	2.06	0.2246
Residual	15.11	4	3.78	--	--
Lack of fit	15.11	3	5.04	--	--
Pure error	0.0000	1	--	--	--
Total	2439.45	9	--	--	--
Model: quadratic; R^2^: 0.9916; R^2^ adjusted: 0.9810; R^2^ predicted: 0.9135; adequate precision: 31.2851; S.D: 2.05; CV: 5.19.					
**Source (CS-BL)**	**Sum of Squares**	**Degree of Freedom**	**Mean Square**	***F*-Value**	***p*-Value**
Model	658.33	5	131.67	283.23	<0.0001 ^a^
A	112.75	1	112.75	242.55	<0.0001
B	544.35	1	544.35	1170.97	<0.0001
AB	0.9409	1	0.9409	2.02	0.2279
A^2^	0.0211	1	0.0211	0.0453	0.8419
B^2^	0.2315	1	0.2315	0.4980	0.5193
Residual	1.86	4	0.4649	--	--
Lack of fit	1.86	3	0.6198	--	--
Pure error	0.0000	1	0.0000	--	--
Total	660.19	9	--	--	--
Model: quadratic; R^2^: 0.9971; R^2^ adjusted: 0.9937; R^2^ predicted: 0.9736; adequate precision: 52.4867; S.D: 0.6818; CV: 2.83.					
**Source (CS-MD)**	**Sum of Squares**	**Degree of Freedom**	**Mean Square**	***F*-Value**	***p*-Value**
Model	1976.70	5	395.34	93.94	0.0003 ^a^
A	1198.03	1	1198.03	284.66	<0.0001
B	690.37	1	690.37	164.04	0.0002
AB	13.32	1	13.32	3.17	0.1498
A^2^	71.52	1	71.52	16.99	0.0146
B^2^	0.1803	1	0.1803	0.0428	0.8461
Residual	16.83	4	4.21	--	--
Lack of fit	16.33	3	5.44	10.89	0.2183 ^b^
Pure error	0.5000	1	0.5000	--	--
Total	1993.53	9	--	--	--
Model: quadratic; R^2^: 0.9915; R^2^ adjusted: 0.9809; R^2^ predicted: 0.9134; adequate precision: 31.2851; S.D: 2.0514; CV: 5.1897.					

## Data Availability

The authors affirm that this paper has the information needed to support the study’s results.

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
