# Peer review of "Use of Insect-Derived Chitosan for the Removal of Methylene Blue Dye from Wastewater: Process Optimization Using a Central Composite Design"

_materials, 2023, doi:10.3390/ma16145049_

Round 1

Reviewer 1 Report

Article entitled Use of insect-derived Chitosan for Azo Dye Removal from Wastewater: Process Optimisation Using Central Composite Design written by Ilham Ben Amor, Hadia Hemmami, Salah Eddine Laouini, Soumeia Zeghoud, Mourad Benzina, Sami Achour, Abanoub Naseef, Ali Alsalme and Ahmed Barhoum and submitted to Materials journal as a draft no materials-2452165 deals with an important issue of ecological sorbents developement for pollutants removal.

The article is in journal’s scope. Therefore, it could be considered for publication in Materials journal. As English is not my native language, I am not able to assess language correctness. However, while reading, I found some statements missing, confusing or unclear. Below I enclose the list of my comments.

Novelty statement should be given at the end of introduction.

Some literature review on current chitosan applications should be given in introduction.

The profile of the Materials journal is, as the name suggests, related to materials and their practical applications. The article lacks an in-depth analysis of the material. It is worth determining the specific surface area, particle size, surface functional groups, and taking pictures eg with SEM. It would be necessary to examine the properties of the material after the process, determine what has changed. The possibility of desorption of MB and reuse of chitosan should be considered.

I do not find obvious errors in the methodology used by the Authors, however, as the Authors write, performing only 9 experiments is very little. Drawing far-reaching statistical/model-based conclusions based on only 9 samples/results is very risky.

The article lacks consideration of the MB removal mechanism. Similarly, its mathematical description is missing. This needs to be supplemented, but it requires more experimental data.

In its current form, the article is not suitable for publication due to numerous deficiencies in the methodology and the method of performing the experiment. However, the tests performed have the potential for publication, and due to the fact that the Authors performed tests on a model MB solution, the gaps can be filled.

Based on my comments and overall impression, I suggest major revision.

Author Response

Reviewer #1:

The article is in the journal’s scope. Therefore, it could be considered for publication in the Materials journal. As English is not my native language, I am not able to assess language correctness. However, while reading, I found some statements missing, confusing or unclear. Below I enclose the list of my comments.

Comment 1: Novelty statements should be given at the end of the introduction.

Reply 1: Novelty has been added

Comment 2

Some literature review on current chitosan applications should be given in the introduction.

Reply 2: Novelty has been added

Chitosan has been investigated for its capacity to remove various organic compounds (dyes, phenolic and pharmaceutical compounds, herbicides, pesticides, drugs, etc.) from wastewater [20], owing to its excellent adsorption capabilities and biodegradability. The effectiveness of chitosan in removing MB dye from wastewater depends on various factors, including the type of chitosan, chitosan dosage, pH, contact time, and dye concentration. Surprisingly, the application of Central Composite Design (CCD) has not been explored for the removal of MB dye from wastewater using chitosan. This study aims to extract chitosan from widespread insects (Pimelia fernandezlopezi (CS-PF), Blaps lethifera (CS-BL), and Musca domestica (CS-MD)) and optimize the processing conditions to achieve efficient MB dye removal from wastewater using CCD. The novelty lies in the utilization of chitosan extracted from insects as an adsorbent for removing methylene blue dye. This approach presents a sustainable and renewable source of chitosan, taking advantage of the abundance and rapid reproduction rates of insects. The study further enhances the novelty by employing the CCD for process optimization, which allows for a systematic exploration of various parameters and their interactions.

Comment 3: The profile of the Materials journal is, as the name suggests, related to materials and their practical applications. The article lacks an in-depth analysis of the material. It is worth determining the specific surface area, particle size, and surface functional groups, and taking pictures eg with SEM. It would be necessary to examine the properties of the material after the process, and determine what has changed. The possibility of desorption of MB and reuse of chitosan should be considered.

Reply 3: SEM images of Chitosan have been added, see Figure 1.

Comment 4: I do not find obvious errors in the methodology used by the Authors, however, as the Authors write, performing only 9 experiments is very little. Drawing far-reaching statistical/model-based conclusions based on only 9 samples/results is very risky.

Reply 4: Many thanks for your comment in future work we will consider increasing the number of experiments.

Comment 5: The article lacks consideration of the MB removal mechanism. Similarly, its mathematical description is missing. This needs to be supplemented, but it requires more experimental data.

Reply 5: A new title has been added that explains the mechanism of dye removal by chitosan:

See Chapter 4. MB removal mechanism

Comment 6: In its current form, the article is not suitable for publication due to numerous deficiencies in the methodology and the method of performing the experiment. However, the tests performed have the potential for publication, and due to the fact that the Authors performed tests on a model MB solution, the gaps can be filled.

Reply 6: Many thanks for confirming that the tests performed have the potential for publication.

Reviewer 2 Report

In this manuscript, the authors studied insect-derived chitosan for azo dye removal from wastewater. This work seems to be useful for this field. However, the following problems should be addressed before further consideration of publication:

1. The title needs revision to be concise.

2. The term abbreviations in abstract/body should be explained at the first time.

3. In the Introduction, the related advances in this field should be added including 10.1021/acsami.1c18435; 10.1002/EXP.20220050; 10.1016/j.apsusc.2021.152165.

4. A scheme can be added for better demonstration of the contents and innovations.

5. All the figures need to be revised in consistent layout/marks to improve the readability.

6. The authors should add some basic characterization results of insect-driven chitosans, to better show its material properties before dye removal evaluation.

7. Format problems in section 2.3 (line 115-129). All the equations should be numbered.

8. Figure 5 is confusing which needs detailed explanation. It is also suggested to be optimized using suitable expressions.

Author Response

Reviewer #2:

In this manuscript, the authors studied insect-derived chitosan for azo dye removal from wastewater. This work seems to be useful for this field. However, the following problems should be addressed before further consideration of publication:

Comment 1: The title needs revision to be concise.

Reply 1: the title changed as recommended.

Comment 2: The term abbreviations in abstract/body should be explained at the first time.

Reply 2: the terms have been explained

Comment 3:  In the Introduction, the related advances in this field should be added including 10.1021/acsami.1c18435; 10.1002/EXP.20220050; 10.1016/j.apsusc.2021.152165.

Reply 3: ref have been added

Comment 4: A scheme can be added for a better demonstration of the contents and innovations.

Reply 4: a graphical abstract and mechanism of dye removal were added to the article.

Comment 5: All the figures need to be revised in a consistent layout/marks to improve readability.

Reply 5: The figure’s layout was improved as recommended

Comment 6: The authors should add some basic characterization results of insect-driven chitosans, to better show its material properties before dye removal evaluation.

Reply 6: SEM images and crystallinity were added to the result section.

Comment 7: Format problems in section 2.3 (lines 115-129). All the equations should be numbered.

Reply 7: All equations in the manuscript have been numbered

Comment 8: Figure 5 is confusing and needs a detailed explanation. It is also suggested to be optimized using suitable expressions.

Reply 8: Expression is improved

In the CCD, the process factors are deliberately manipulated based on a predetermined plan, often a response surface design, to analyze their impact on the response variable of interest. This design includes central points and star points, which facilitate the estimation of linear, quadratic, and interaction effects of the process factors on the response variable. Visualizing the data using 2D and 3D plots helps in understanding the relationships between the variables and the response. Figure 5 presents contours and response surface plots, offering a graphical representation of the variations in methylene blue (MB) removal using Cs extracted from three different insects. The contours demonstrate that higher MB removal is achieved at higher MB concentrations and longer reaction times. Notably, CS-BL exhibits the highest MB removal efficiency, followed by CS-MD and then CS-PF, indicating that the extraction of chitosan from insects, particularly CS-BL, shows promising potential for dye removal applications.

Reviewer 3 Report

Summary and general comments

In this study, the authors investigate the use of insects-derived chitosan as adsorbent for the removal of methylene blue from the aqueous solution. The authors studied only the adsorbate concentration and agitation time and fed the data into response surface methodology software, which explore the relationship between parameter.

The Introduction and the Methodology sections are currently lacking and should be improved. Please see the specific comment below for details.

Specific Comments

1.   Title Instead of using the term Azo, I recommend the authors to change it to Methylene blue.

2.   Abstract: The biological name of the genus and species should be in italics. Authors should do the same in the main text.

3.   The term Azo is too general. I suggest the author to label the dye class of methylene blue, which is a cationic thiazine dye.

4.   Abstract. CCD should be mentioned in full at the first appearance

5.   Abstract lines 29-30. Avoid mentioning % removal without giving the initial adsorbate concentration. Otherwise, use the adsorption capacity at equilibrium;

6.   The keyword section should include methylene blue and Central Composite 3 Design.

7.    The introduction should include some descriptions of the insects, why are they chosen for their chitosan source. What are their attractive features as adsorbents?If these insects have common names, they should be included. This is to provide sufficient background so the readers do not need to look for external sources.

8.   Lines 55-57. The choice of adsorbents is very narrow. I suggest the authors to include biochar, biomass source of high availability, etc. here are a few recommendations: doi.org/10.1155/2021/5932222
doi.org/10.1155/2022/8245797

9.   Line 103- 114. The equations should be done properly using equation builder, and assign equation number.

10.             How was the ash content determined and at what temperature, using what furnace?

11.             How was the moisture content, WBC and FBC carried out? Brief experimental procedures must be included. This is to ensure that the experiment can be reproduced.

12.             What is the nature of the extracted chitosan? What are their particle size.

13.             The detailed experimental procedure of the batch adsorption experiment should be provided. How was the adsorption process carried out, the experimental condition (pH, agitation time, adsorbent dosage, adsorbate concentration, etc.), What are the volume of adsorbate? how was the adsorbent separated after the adsorption process, and how was the methylene blue dye determined and used using what machine, and at what wavelength. All of this information must be included.

14.             Was the effect of pH investigated?

15.             The equations on lines 250-252 should be numbered.

16.             The expression of the unit in the main text and figures and table should be consistent. mgL-1 was used in the text, while in the figure is mg/l

Inconsistencies in units used. see specific comment to authors for details

Author Response

Reviewer #3:

In this study, the authors investigate the use of insects-derived chitosan as adsorbent for the removal of methylene blue from the aqueous solution. The authors studied only the adsorbate concentration and agitation time and fed the data into response surface methodology software, which explore the relationship between parameter.

The Introduction and the Methodology sections are currently lacking and should be improved. Comment 1: Title Instead of using the term Azo, I recommend the authors to change it to Methylene blue.

Reply 1: the title changed as recommended.

Comment 2

Abstract: The biological name of the genus and species should be in italics. Authors should do the same in the main text.

Reply 2: All manuscripts have been changed

Comment 3: The term Azo is too general. I suggest the author to label the dye class of methylene blue, which is a cationic thiazine dye.

Reply 3: the AZO dye has been changed to methylene blue

Comment 4: Abstract. CCD should be mentioned in full at the first appearance

Reply 4: It was added in Abstract: Central Composite Design (CCD)

Comment 5: Abstract lines 29-30. Avoid mentioning % removal without giving the initial adsorbate concentration. Otherwise, use the adsorption capacity at equilibrium;

Reply 5: The percentage of dye removal was added with the initial dye concentration

Comment 6. The keyword section should include methylene blue and Central Composite 3 Design.

Reply 6: Words have been added to the keywords

Comment 7: The introduction should include some descriptions of the insects, why are they chosen for their chitosan source. What are their attractive features as adsorbents? If these insects have common names, they should be included. This is to provide sufficient background so the readers do not need to look for external sources.

Reply 7: Added in the introduction

Comment 8: Lines 55-57. The choice of adsorbents is very narrow. I suggest the authors to include biochar, biomass source of high availability, etc. here are a few recommendations: doi.org/10.1155/2021/5932222

doi.org/10.1155/2022/8245797

Reply 8: It was added to a paragraph and supported by added references

Comment 9: Line 103- 114. The equations should be done properly using equation builder, and assign equation number.

Reply 9: All equations were done correctly using the equation builder, and set the number of all equations in the manuscript

Comment 10: How was the ash content determined and at what temperature, using what furnace?

Reply 10: The resulting chitosan was vacuum-dried for 4 hours at 650 °C.

See 2.2 Method of extracting chitosan from different insects

Comment 11: How was the moisture content, WBC and FBC carried out? Brief experimental procedures must be included. This is to ensure that the experiment can be reproduced.

Reply 11: Experiments are explained and added to the title

See 2.2 Method of extracting chitosan from different insects

The WBC and FBC were calculated using the formulas:

, and  [23]

Comment 12

 What is the nature of the extracted chitosan? What are their particle size?

Reply 12: SEM images were added to the article with description, see Figure 1.

Comment 13: The detailed experimental procedure of the batch adsorption experiment should be provided. How was the adsorption process carried out, the experimental condition (pH, agitation time, adsorbent dosage, adsorbate concentration, etc.), What are the volume of adsorbate? how was the adsorbent separated after the adsorption process, and how was the methylene blue dye determined and used using what machine, and at what wavelength. All of this information must be included.

Reply 13: this experimental details described in: 2.3 Experimental design

For the equilibrium experiments, measured amounts of Cs were placed in 1 g flasks containing 30 ml of aqueous MB dye solution with varying concentrations ranging from 6.39 to 12.79 mg/L. The duration of the experiments varied from 10 to 120 min, depending on the specific time interval. Throughout the experiments, the temperature was maintained at 25°C, and the mixture was shaken using an orbital shaker at 150 rpm. To halt the adsorption process, a centrifuge operating at a speed of 6000 rpm was utilized to separate the chitosan from the solution. The initial and final concentrations of the MB dye were measured using a UV-Vis spectrophotometer (UV-2450, Shimadzu) at 663 nm, allowing for the determination of the adsorption efficiency for each experiment. The pH of the experimental solution was adjusted at pH=7 using 1 N HCl and 1 N NaOH as needed. 

Comment 14: Was the effect of pH investigated?

Reply 14: No, The pH of the experimental solution was adjusted at pH=7 using 1 N HCl and 1 N NaOH as needed.

Comment 15: The equations on lines 250-252 should be numbered.

Reply 15: All equations have been numbered in the manuscript

Comment 16: The expression of the unit in the main text and figures and table should be consistent. mgL-1 was used in the text, while in the figure is mg/l

Reply 16: The required change was applied to the manuscript (mg/l)

Reviewer 4 Report

Reviewer’s Comments

The submitted manuscript is nicely written and well presented. However, it can be further improved by considering the following:

1.       I think there is a typo error in line 30 in the abstract where the authors have written CS-PF twice in the same sentence.

2.       The sentence beginning in line 30 “The highest MB removal” ends with a closed bracket which I think is another type error. Also, the same sentence needs to be rephrased.

3.       Write the full form of abbreviation “CCD” followed by the abbreviated form in line 28 in the abstract.

4.       The prepared adsorbent is listed as CS-BL, CS-PF and CS-MD in the abstract. I think, the same order should be followed throughout the manuscript even in Tables. The order is followed in Table 3 but not in Table 2. This will make it easy for the readers to navigate and follow the trends and observations reported.

5.       The authors claim that the highest MB removal efficiency was observed for CS-BL and lowest for CS-MD. The explanation for such results should be more detailed. Is it due to high value of DD or it is due to the higher moisture content in CS-BL or is it due to the higher water binding capacity of CS-BL.

6.       Methodology for the various characteristics presented in Table 2 should be clearly explained.

7.       In line 167 in the Results and Discussion section, it is stated that characterization is important but very limited characterization results are presented. Characterization such as IR, XRD, SEM, BET surface area measurements are highly recommended that could easily prove the obtained results.

8.       Section 2.3 – Experimental Design – the font should be justified.

9.       The text in line 129 in the experimental design section “Experiments shown in Table 1” should be deleted. I think, there is no need for it.

10.   Typo error in line 146 – “Chitosan in 1 g flasks” should be modified.

11.   References should be cited in sequential order. For example, references 14, 15 on page 2 line 69 followed by reference 17 on page 3 line 97. Similarly, references 26 and 27 is followed by reference 39 on page 5 line 201. Please check the consistency of all cited references.

12.   Explanation on page 5 (Results and discussions) is mostly description rather than discussions. Please try to focus on describing your results.

13.   Page 6 – line 234, “In previous studies, it was found” needs some clarification and reference.

The overall, English language of the submitted manuscript should be further improved.  

Minor English language improvements are needed.

Author Response

please find attached the responses to the reviewer's comments 

Round 2

Reviewer 1 Report

This is my second review of this article. Corrected version is better than the first one. The Authors answered my questions and made many corrections. However, the most important things still need corrections and supplementation. I still have doubts about the characterization of the material. Although SEM photos have been added, the parameters of the parameter still remain unknown. It is difficult to talk about the preparation of the material without knowing its characteristics. Moreover, there are no characteristics of the material after its use. There are no considerations of reuse, etc. In conclusion, an in-depth material analysis should be performed thoroughly and some of the experiments in the laboratory repeated which have not been done. The second doubt raised in my first review is the very small number of experiments (9). The Authors did not address this in any way. Therefore, I am repeating the question, at the same time expanding it asking Authors for modeling the minimum number of experiments that would allow for the calibration and correct use of such statistical models.

Author Response

We have carefully considered all comments and here is a point-by-point response to the reviewers’ comments

Reviewer #1:

This is my second review of this article. Corrected version is better than the first one. The Authors answered my questions and made many corrections. However, the most important things still need corrections and supplementation. I still have doubts about the characterization of the material.

Comment 1: Although SEM photos have been added, the parameters of the parameter still remain unknown. It is difficult to talk about the preparation of the material without knowing its characteristics. Moreover, there are no characteristics of the material after its use. There are no considerations of reuse, etc. In conclusion, an in-depth material analysis should be performed thoroughly, and some of the experiments in the laboratory repeated which have not been done.

Reply 1: The prepared chitosans were fully characterized, based on three repeats; see Table 2 and SEM images

 Table 2. Properties of chitosan extracted from Blaps lethifera (CS-BL), Pimelia fernandezlopezi (CS-PF), and Musca domestica (CS-MD). The values are based on a three measurements at least.

Characteristics of chitosan

Sources chitosan extracts

CS-BL

CS-PF

CS-MD

Yield (Y)

41.5±0.5%

50.3±0.3%

57.9±0.2%

Moisture content (MC)

17.2±0.2%

14.3±0.3%

7.8±0.1%

Ash contents (AC)

1.6±0.1%

2.1±0.1%

8.2±0.2%

Water binding capacity (WBC)

515.5±6.5%

287.1±5.8%

301.1±4.3%

Fat binding capacity (FBC)

433.4±11.3%

296.8±14%

455.2±13.2%

Degree of deacetylation (DD)

87.1±0.2%

88.3±0.1%

84.1±0.3%

Crystallinity Index

84±0.1%

73±0.4%

81±0.2%

The values are based on three measurements at least and the fitting data given in the text as well as in the The results of ANOVA are shown in Table 3. Determining the relationship between the dependent factors and the response (MB distance) and the values of R2adj, R2, Adeq. Precision, lack of fit, F-value, CV, and SD. High F-values and low p-values represent the significance of the model performed by the test of ANOVA[38]. The R2 coefficient, which measures the fit of the model and is defined as the ratio of the specified variable to the total variance, was convincingly good for the model (CS-PF, R2: 0.9916; R2 Fit: 0.9810; CS-BL, R2: 0.9971; R2 Fit: 0.9937; CS-MD, R2: 0.9915; R2 Fit: 0.9809).

The limitation of this study was discussed in chapter 4. MB removal mechanism “In this particular study, the focus was primarily on investigating the adsorption capacity of chitosan for methylene dye removal, and the regeneration process of chitosan was not explored. To reuse chitosan for subsequent dye removal, regeneration becomes necessary, which involves the desorption of the adsorbed dye molecules from the chitosan surface. Several regeneration methods exist, such as altering the solution's pH, employing organic solvents, or using specific desorbing agents, with the choice dependent on the adsorption conditions and desired regeneration efficiency. However, the regeneration aspect was not considered within the scope of this study, and further research or experimentation would be required to evaluate the effectiveness and viability of chitosan regeneration for subsequent dye removal.”

Comment 2: The second doubt raised in my first review is the very small number of experiments (9). The Authors did not address this in any way. Therefore, I am repeating the question, at the same time expanding it asking Authors for modelling the minimum number of experiments that would allow for the calibration and correct use of such statistical models.

Reply 2: We would like to provide further clarification based on your query. Our study consisted of a total of 10 experiments (see, Table). To enhance the efficiency of our experimental design, we identified and eliminated a repetitive experiment that was already accounted for in our plan. This decision was made to minimize redundancy and ensure the reliability of our results. In our aim to strike a balance between reducing the number of experiments and obtaining robust outcomes, we strategically integrated two central points into our experimental design. This approach allowed us to optimize the number of experiments while maintaining statistical significance and reducing unnecessary repetition. While the determination of the minimum number of experiments necessary for accurate calibration and correct use of statistical models in Central Composite Design (CCD) is crucial, our selection of the number of experiments was primarily guided by practical considerations and the resources available for our study. However, we acknowledge the importance of further modeling and analysis to determine the minimum number of experiments required to achieve precise calibration. This will enhance the validity and reliability of future studies.

Reviewer 2 Report

I have checked all the content revisions. The revised manuscript has improved most issues, yet the following problems still need to be addressed:

1. For better readability, Figure 1 of several SEM images should be in the same size and layout. Clear scale bars are also needed.

2. In the Introduction, references should be added to show the related advances in this field including 10.1021/acsami.1c18435; 10.1002/EXP.20220050, which are also suggested to be useful.

3. All the subgraphs need to be marked (e.g., a, b, c or a1, a2, a3). Figure 1 and Figure 6 can be revised in this case.

4. The whole manuscript should be checked thoroughly considering expression errors. For example, “Cs” in line 167, page 4. Generally, “1 N” can also be replaced with “1 M” or “1 mol/L”. All the physical quantities in the text should be expressed in italics. etc...

Author Response

Reviewer #2:

I have checked all the content revisions. The revised manuscript has improved most issues, yet the following problems still need to be addressed:

Comment 1: For better readability, Figure 1 of several SEM images should be in the same size and layout. Clear scale bars are also needed.

Reply 1: The figures have been modified

Comment 2:  In the Introduction, references should be added to show the related advances in this field including 10.1021/acsami.1c18435; 10.1002/EXP.20220050, which are also suggested to be useful.

Reply 2: refs have been added

Comment 3: All the subgraphs need to be marked (e.g., a, b, c or a1, a2, a3). Figure 1 and Figure 6 can be revised in this case.

 Reply 3: Figure 1 and Figure 6 have been revised as recommend

Comment 4: The whole manuscript should be checked thoroughly considering expression errors. For example, “Cs” in line 167, page 4. Generally, “1 N” can also be replaced with “1 M” or “1 mol/L”. All the physical quantities in the text should be expressed in italics. etc...

 Reply 4: thanks for your valuable comments, these changes are related to the MDPI format and proofreading stage. As authors, we do not know the MDPI format but will be in touch with the MDPI proofreading team.

Reviewer 3 Report

Most of the comments raised previously are amended.

Author Response

many thanks for your kind support. 

Reviewer 4 Report

The authors have made significant improvements to the manuscript and all the recommended corrections are being incorporated.

Round 3

Reviewer 1 Report

This is my third rewiev of this article. The Authors answered my comments, but once again their answer is unsatisfactory. The profile of the Materials journal is related to the characterization of new materials and their practical application. The Authors really use interesting material. However, it is very important to accurately characterize the material. This is missing. By the SEM alone and the data from the table do not allow to draw a conclusion as to why there are such large differences between the materials. The number of experiments is very limited. The Authors claim to have performed 10 experiments. The Authors use advanced statistical models and methods. However, it is worth considering what is the minimum number of experiments that allows the research to be considered correct. With such volatility of parameters and a large number of materials used, I have serious doubts about the reliability.

Reviewer 2 Report

I have checked all the revisions, and recommend it for publication in Materials.